Habitat creation and biodiversity maintenance in mangrove forests: teredinid bivalves as ecosystem engineers

Hendy Ian W. ian.hendy@port.ac.uk
Michie Laura
Taylor Ben W.
Institute of Marine Sciences, The University of Portsmouth , UK
Esler Karen
Electronic publication date: 2014 Sep 25
Publication date: 2014
Volume: 2
Electronic Location ID: e591
Received 2014 Aug 11; Accepted 2014 Sep 3
Copyright: © 2014 Hendy et al.
Copyright year: 2014
Copyright holder: Hendy et al.
License: This is an open access article distributed under the terms of the Creative Commons Attribution License, which permits unrestricted use, distribution, reproduction and adaptation in any medium and for any purpose provided that it is properly attributed. For attribution, the original author(s), title, publication source (PeerJ) and either DOI or URL of the article must be cited.
License URL: https://creativecommons.org/licenses/by/4.0/

Keywords: Mangrove wood, Teredinid tunnels, Refuge, Biodiversity, Niche creation, Cryptic fauna, Nursery

Funding: Operation Wallacea This study was funded by Operation Wallacea. The funders had no role in study design, data collection and analysis, decision to publish, or preparation of the manuscript.

==============================
Substantial amounts of dead wood in the intertidal zone of mature mangrove forests are tunnelled by teredinid bivalves. When the tunnels are exposed, animals are able to use tunnels as refuges. In this study, the effect of teredinid tunnelling upon mangrove forest faunal diversity was investigated. Mangrove forests exposed to long emersion times had fewer teredinid tunnels in wood and wood not containing teredinid tunnels had very few species and abundance of animals. However, with a greater cross-sectional percentage surface area of teredinid tunnels, the numbers of species and abundance of animals was significantly higher. Temperatures within teredinid-attacked wood were significantly cooler compared with air temperatures, and animal abundance was greater in wood with cooler temperatures. Animals inside the tunnels within the wood may avoid desiccation by escaping the higher temperatures. Animals co-existing in teredinid tunnelled wood ranged from animals found in terrestrial ecosystems including centipedes, crickets and spiders, and animals found in subtidal marine ecosystems such as fish, octopods and polychaetes. There was also evidence of breeding within teredinid-attacked wood, as many juvenile individuals were found, and they may also benefit from the cooler wood temperatures. Teredinid tunnelled wood is a key low-tide refuge for cryptic animals, which would otherwise be exposed to fishes and birds, and higher external temperatures. This study provides evidence that teredinids are ecosystem engineers and also provides an example of a mechanism whereby mangrove forests support intertidal biodiversity and nurseries through the wood-boring activity of teredinids.

Introduction

Mangrove ecosystems have long been considered as low bio-diverse habitats (Duke, Ball & Ellison, 1998; Alongi, 2002), especially when compared with other tropical marine ecosystems, for example, coral reefs (Connell, 1978; Knowlton et al., 2010). However, mangrove forests provide a variety of niches to creatures that depend upon these coastal ecosystems (Nagelkerken et al., 2008). Animals commonly described from mangrove forests are found in a broad range of different biomes, such as aquatic sesarmid crabs, which break down and recycle much of the leaf litter (Robertson, 1990), and terrestrial beetles that process large woody debris (LWD) in the high- to mid-intertidal areas of the forests (Feller, 2002). Thus, mangrove forests host organisms commonly found in terrestrial and aquatic habitats (Nagelkerken et al., 2008). Principally, within the mangrove environment, there are three main substrata that fauna are able to exploit; sediments (Kristensen, 2007), root structures (Ellison & Farnsworth, 1990; Ellison, Farnsworth & Twilley, 1996) and LWD (Cragg & Hendy, 2010; Hendy et al., 2013).

Studies of tropical habitat structure have shown that the structural complexity maintains the greatest level of biodiversity (Gratwicke & Speight, 2005; Fuchs, 2013). Mangrove forests provide a wide range of niches maintained by substrata such as mangrove roots, which maintain a high level of biodiversity (Gratwicke & Speight, 2005). The complexity of root structures provides cover and protection for small and juvenile fish communities (Ronnback et al., 1999; Correa & Uieda, 2008; Wang et al., 2009) that decrease their risk of becoming predated upon (Verweij et al., 2006; Tse, Nip & Wong, 2008; MacDonald, Shahrestani & Weis, 2009). The structural heterogeneity provided by the roots may either impede the movement of hunting predatory fish or the prey-fish are able to reduce their visibility by using the roots to hide behind (Laegdsgaard & Johnson, 2001; Kruitwagen et al., 2010)—therefore the diversity of animals and abundance of individuals are largely considered to be attributed to deterministic factors such as habitat complexity (Syms & Jones, 2000). Indeed, habitat complexity is one of the most important factors structuring faunal assemblages —acting as a decoupling mechanism in predator–prey interactions (Firstater et al., 2011; Kovalenko, Thomaz & Warfe, 2012).

Although mangrove roots provide a nursery habitat, little is known about the mangrove faunal communities relying upon fallen wood as a habitat; in particular wood that has been attacked by teredinid bivalves. Teredinids create many tunnels in LWD (Filho, Tagliaro & Beasley, 2008). When the teredinids die, the tunnels may support biodiversity when vacant, for animals to exploit (Cragg & Hendy, 2010; Hendy et al., 2013). This means that LWD may provide an important ecosystem service as a bio-diverse micro-habitat within intertidal areas (Storry et al., 2006). It is known that when trees fall into freshwater ecosystems they attract a high level of biodiversity (Roth et al., 2007). The riparian zone is important as this habitat sustains many aquatic animals. Thus, LWD serves as an important link between terrestrial and aquatic ecosystems (Roth et al., 2007).

This study aims to investigate the ecological role of teredinid bivalves in creating niches—tunnels for fauna, and if a greater number of tunnels maintain greater abundances of animals and numbers of species within LWD from Indonesian mangrove ecosystems. Thus, the engineered internal structural complexities within dead wood created by teredinid tunnels were quantified and measured, and then correlated with counts of animal abundance and number of species—to determine if increases of habitat complexity created by teredinids can enhance animal abundance and numbers of species. Teredinids may then be considered as allogenic ecosystem engineers. Such organisms modify the environment by transforming living or non-living material (e.g., LWD) from physical state to another, via mechanical processes to maintain and create habitat (Jones, Lawton & Shachak, 1994).

To increase biodiversity the engineering species, in this case, teredinids must create conditions not present elsewhere in the landscape, and other animals must be able to live in the engineer-created patches (Wright, Jones & Flecker, 2002). Thus, this study tested the hypotheses that:

1. the tunnels created by teredinids, when vacant are exploited by other fauna

2. with a greater number of teredinid tunnels there will be a greater level of animal diversity within fallen wood

3. fallen wood provides a refuge for mangrove fauna, from extreme environmental conditions due to cooler internal temperatures

Materials and Methods

Assessing the effect of teredinid tunnelling on LWD in mangrove forests

The abundance of animals and species within teredinid tunnels was estimated from five mangrove forests Langira, Kaluku, Loho, One Onitu and the Gili forest in East Sulawesi, Indonesia (05°12′–06°10′S, 123°20′–124°39′E). Within each forest, LWD samples were collected from five transects extending from the strandline out to the seaward edge. Each transect was 4 m wide, and transects were between 50 and 100 m apart. For details of transect lengths and total numbers of LWD samples collected, see Table 1. All wood was measured and a sample from every piece of wood within the transects that fitted the criteria for LWD (>20 mm diameter) was removed for examination, and measured. From each mangrove site, twenty LWD samples were chosen at random to compare the abundance of animals, numbers of species and percentage surface area of tunnels. The proportion of sampled wood from the total volume of fallen wood in each of the transects from each mangrove forest ranged from 0.4 to 5.9% of the total volume (Table 2B). Counts of animals removed from within the tunnels in each sample of wood were estimated to 1 litre and were expressed as numbers per litre of wood.

Table 1 Transect lengths and wood collection details.

The range of lengths (metres) of five transects extending from the strandline, and out to the fringing edge from each mangrove forest locality, combined with the total number of wood samples collected from the five transects in each mangrove forest.

Site	Transect length (m)	Number of LWD samples	
Langira	340 to 440	70	
Kaluku	25 to 60	20	
Loho	60 to 160	44	
One Onitu	80 to 110	32	
Gili	70 to 100	30	

Table 2 The range of animals in teredinid tunnels and mangrove forest details.

(A) Animals removed from teredinid tunnels in 20 samples of wood, from each of the five mangrove forest sites. (B) details of the five mangrove forest sites, with the total area of each forest (hectares), the total area surveyed of the combined five transects (hectares), the total volume of fallen wood found in the five transects (m3), the % of wood used to quantify animals and teredinid attack from the total volume of wood within the five transects, and the total number of species and abundance of animals removed from 20 wood samples from each mangrove forest locality.

A	
				SITES	
Phylum	Class	Family	Species	Langira	Kaluku	Loho	One Onitu	Gili	
Platyhelminthes	Rhabditophora	Gnesiocerotidae	Styloplanocera sp. A		*	*			
Nemertea	Enopla	Prosorhochmidae	Pantinonemertes sp. A		*	*			
Mollusca	Bivalvia	Isognomonidae	Isognomon ephippium	*	*				
		Mytilidae	Xenostrobus sp. A	*	*	*			
	Gastropoda	Assimineidae	assimineid sp. A		*	*	*		
		Cerithiidae	Clypeomorus sp. A	*					
		Columbellidae	Pseudanachis basedowi	*					
		Ellobiidae	Pythia sp. A					*	
		Marginellidae	marginellid sp. A		*	*	*		
		Mitridae	Strigatella sp. A		*				
		Muricidae	Thais gradata				*		
		Nassariidae	Nassriuas sp. A	*					
		Onchidiidae	Onchidium nigram		*				
		Pyramidellidae	pyramidellid sp. A		*				
	Polyplacophora		chiton sp. A			*			
	Cephalopoda		octopod sp. A	*			*		
Sipuncula	Sipunculidea	Phascolosomatidae	Phascolosoma arcuatum	*	*				
Annelida	Polychaeta	Amphinomidae	amphinomid sp. A	*	*	*		*	
		Eunicidae	eunicid sp. A	*	*	*	*		
		Nereididae	nereid spp.	*	*	*	*		
		Terebellidae	terebellid sp. A	*	*	*	*		
Arthropoda	Arachnida	Desidae	Desis martensi	*	*	*	*		
	Chilopoda	Cryptopidae	centipede sp. A		*			*	
	Insecta	Cerambycidae	cerambycid sp. A	*	*		*	*	
		Gryllidae	Apteronemobius asahinai					*	
	Malacostraca	Alpheidae	Alpheus sp. A	*	*	*			
		Atyidae	Caridina propinqua	*	*			*	
		Diogenidae	Diogenes sp. A	*	*				
		Grapsidae	Metapograpsus spp.	*	*	*	*	*	
		Xanthidae	xanthoid spp.	*	*	*	*	*	
		Cirolanidae	Cirolana sp. A	*	*	*	*	*	
		Talitridae	Microrchestia sp. A	*	*	*	*	*	
Chordata	Actinopterygii	Blenniidae	blennie sp. A				*		
		Muraenidae	Gymnothorax richardsoni	*					
		Ptereleotridae	Parioglossus interruptus	*	*	*			
	Ascidiacea		tunicate sp. A		*		*		
	
B	
Site	Mangrove
area (ha)	Mangrove area
surveyed (ha)	LWD volume in
transects (m3)	% of LWD
analysed	Total
species	Total
abundance	
Langira	60	1	3.5	0.4	22	173	
Kaluku	0.5	0.1	0.22	5.9	26	850	
Loho	3.1	0.2	0.75	2.4	17	216	
One Onitu	1	0.2	1.5	1	15	249	
Gili	1	0.1	1	1	10	132	
Notes.

* Present in that mangrove forest.

The LWD samples were carefully broken apart and all animals were collected and identified. The percentage of teredinid tunnels was used to categorise the level of teredinid attack, by measuring the percentage of the surface occupied by teredinid tunnels across the longitudinal section of each LWD sample using the digital analysis package, Image Tool Version 3.00 (The University of Texas Health Science Centre at San Antonio). Within each mangrove forest, the substratum type that each LWD sample was collected from was noted. Ground water salinities were measured using a Bellingham and Stanley E-Line Aquatic hand-held refractometer, and the distance from the land (strandline) of each LWD sample was recorded. To calculate emersion time within each mangrove forest, at five metre intervals the level of high tide was marked on mangrove trees using high visibility string. This was repeated from the strandline and extending out to the seaward edge. At low tide, the distance from the substratum to the mark on the tree was measured and then subtracted from the height of high tide as given in the Indonesian tide tables. Emersion times were estimated by relating their tidal height to data in regional tide tables.

The field study was agreed and approved by Operation Wallacea under permit 04/TKPIPA/FRP/SM/IV/2011.

In situ internal wood and ambient air temperature measurements

A calibrated thermocouple thermometer (Oakton WD-35427-00 Temp 10 Type J) was used to measure internal temperatures of 27 in situ teredinid-attacked logs and outside air temperatures in the Langira mangrove forest. The thermometer had two temperature probes: one probe was placed upon the wood surface and the other placed in the centre of the wood. The wood was very quickly split open and the probe placed into the centre before closing the split section. After five minutes of placing both probes in position, the temperature was recorded. A sample of wood from each piece of wood that temperatures were recorded from was taken back to the laboratory, and all animals within the vacant teredinid tunnels within the wood sample were removed and counted.

Statistical data analysis

The univariate and non-parametric multivariate techniques of the distance-based linear modelling package (DistLM) contained in PRIMER 6.1 (Plymouth Routines in Multivariate Ecological Research) were used to explore the animal abundance and number of species removed from vacant teredinid tunnels in wood across five mangrove forests, and tested against environmental variables: salinity, substrate type (mud, sand, and calcareous mud), distance from land (D.F.L), percentage surface area of teredinid attack, mangrove area and size of wood sample. DistLM was employed to verify relationships between the abundance of fauna and number of species removed from teredinid tunnels across all sites with the environmental variables. DistLM produces a marginal test, which assesses the variation each predictor (environmental variable) has on its own, and a sequential test, assessing the variation of all the environmental variables (McArdle & Anderson, 2001). The most parsimonious model was identified using the selection criterion R2. Distance-based redundancy analyses (dbRDA) were used for visualizing the results as an ordination, constrained to linear combinations of the environmental variables. The DistLM was based on abundance and environmental data with 4999 permutations.

A 1-way ANOVA was used to test for differences; of the number of species and abundance of animals in wood with different percentage surface areas of teredinid tunnels; site-specific differences of the percentage surface area of teredinid tunnels; and emersion times between mangrove sites. Tukey’s post-hoc pairwise comparison tests separated values into statistically distinct subsets for ANOVA. All data were checked for normality, residuals were inspected to ensure that assumptions for ANOVA were not compromised. Regression analyses were used to test for relationships of animal abundance and numbers of species within vacant teredinid tunnels with environmental factors: distance from the land, volume of LWD, percentage surface area of teredinid tunnels and ambient air- and within wood-temperatures. A Pearson Correlation was used to test for relationships with internal wood cooling and the abundance of fauna within wood. A Paired t-test was used to determine temperature differences within wood and ambient air temperatures. Count data were square root transformed and all percentage data were arcsine transformed. Statistical analyses were performed using MINITAB (MINITAB Inc, version 13.20).

Results

In total, 36 genera, amounting to 1,621 individuals were found in vacant teredinid tunnels inside wood, which consisted of 7 phyla across the five mangrove forest localities (Tables 2A and 2B).

The kinds of animals found inhabiting teredinid tunnels in LWD samples were diverse, and ranged from terrestrial species, such as coleopteran larvae and crickets (Insecta), intertidal species including reef spiders, Desis martensi (Arachnida) and mussels (Bivalvia) and aquatic species including moray eels, Gymnothorax richardsonii (Actinopterygii) and octopods (Cephalopoda). In addition, different stages of animal development were found—as juveniles of many species were also found in the teredinid tunnels (Fig. 1).

Figure 1 A range of animals within teredinid tunnels.

A range of animals each removed from teredinid tunnels in wood. (A) a developing baby octopus. (B) The Reef Spider, Desis martensi, removed from within its tunnel. Further evidence of the teredinid tunnel nursery-function: the desid has an egg-sac below its abdomen. (C) The exposed tentacles of an octopus with egg-sacs. (D) The ventral view of a cirolanid isopod. Note the large egg-sac almost covering the pereopods. (E) a megalopa (juvenile) spider crab, and (F) a Richardson’s Moray eel found within a teredinid tunnel.

One variable, percentage surface area of teredinid attack, was responsible for explaining 28% of the similarity with the abundance of animals and numbers of species removed from tunnelled wood across all mangrove sites (Figs. 2A–2C, DistLM marginal test, F = 74.2, p = < 0.001). No significant correlations were found with the abundance of animals and numbers of species with substrate type, LWD sample volume (L vol), site area, distance from land (D.F.L) and salinity (Fig. 2B, sequential test, p = > 0.05). The most parsimonious model for the five sites explained 31% of the variation, with percentage surface area of teredinid attack, again explaining 28% of the similarity across all sites (DistLM sequential test, F = 74.2, p = < 0.001).

Figure 2 Multivariate ordinations of animal abundance, numbers of species and environmental variables across five mangrove forests.

Distance-based redundancy analysis (dbRDA) expressed as ordinations. (A) the variation of teredinid-attacked wood samples analysed from five mangrove sites in, relation to; (B) measured environmental variables: teredinid-attack (attack), site area, distance from land (D.F.L), substrate type, volume of wood sample (L vol) and salinity. The strongest relationship explaining the scatter of wood samples is correlated with teredinid attack, and (C) a strong relationship is found with the number of species and abundance of animals found in the wood samples with teredinid attack.

No difference was found with the abundance of animals in wood when tested with salinity, increasing distance from the land and with the total area of the mangrove forest (Multi-regression, p = > 0.05). However, a significant difference was found with greater volumes of LWD when correlated with greater numbers of species (F1,78 = 9, p = < 0.01), nonetheless a low relationship was found, R2 (adj) = 10%, although no significant difference was found with greater volumes of LWD when correlated with abundance of animals (F1,78 = 3.5, p = > 0.05, R2 (adj) = 3%). The strongest relationship with best fit and significance for the numbers of species and abundance of animals in LWD was the percentage surface area of teredinid tunnels (Fig. 3A, F1,78 = 35.6, p = < 0.001, R2 (adj) = 30.5%, and Fig. 3B, F1,78 = 33.8, p = < 0.001, R2 (adj) = 29.4% respectively).

Figure 3 Differences of the abundance of and numbers of species in wood categorised by the percentage surface area of teredinid tunnels.

Regression analyses and means of species and abundance of animals removed from wood exposed to different levels of teredinid attack (numbers of teredinid tunnels expressed by the percentage cross-sectional surface area of tunnels in each wood sample). (A) the number of different species (p = < 0.001), and (B) the total abundance of animals (p = < 0.001), within per litre volume of LWD with different levels of teredinid-attack. (C) the mean number of different species (p = < 0.001), and (D) mean abundance of animals (p = < 0.001) within per litre volume of LWD categorised by percentage surface area of teredinid tunnels. *Note: for each of the four categories of percentage surface area of teredinid tunnels (0–19.9%, 20–39.9%, 40–59.9% and >60%), twenty items of LWD were used (letters above the bars = Tukey’s pairwise post-hoc comparisons, mean ± SE).

The lowest number of species was found in wood with percentage surface area of teredinid tunnels ranging from 0 to 19.9%, with 1.5 ± 0.3 species. The greatest number of species was found in wood with more than 60% surface area of teredinid tunnels, with 10.1 ± 1.0 species (Fig. 3C, 1-way ANOVA, F3,76 = 13.9, p = < 0.001, mean ± SE). The same significant pattern with the abundance of animals was found in LWD with different levels of percentage surface area of teredinid tunnels, with 2.3 ± 0.4 animals in wood with 0–19.9% surface area of tunnels, compared with 27 ± 6.9 animals in wood with more than 60% surface area of tunnels (Fig. 3D, F3,76 = 14, p = < 0.001, mean ± SE).

Although a strong correlation with the percentage surface area of teredinid attack was found with the abundance of animals and number of species in tunnelled wood across all mangrove sites, differences of the percentage surface area of teredinid attack were found between the mangrove sites (Fig. 4A, 1-way ANOVA, F4,95 = 3.6, p = < 0.01, mean ± SE). The percentage surface area of teredinid attack between three of the mangrove forests was statistically indistinguishable, while the percentage surface area of teredinid attack in the Gili forest was significantly lower than Langira, Loho and Kaluku, 8.7% ± 3.5%, 16.5% ± 4%, 23.9% ± 5.3% and 32.3% ± 5.6% respectively. The greatest degree of teredinid-tunnelling was found in LWD samples from the Kaluku mangrove forest.

Figure 4 Mangrove forest differences of teredinid tunnels in fallen wood, emersion time and animals within tunnels.

The effect from vacant teredinid tunnels and emersion on five different mangrove forests. (A) the percentage of cross-sectional surface area of teredinid tunnels measured in wood (n = 20) from each of the five mangrove forest localities (p = < 0.01). (B) emersion times (hd−1) within each of the five mangrove forests (p = < 0.001). (C) the abundance of animals removed and counted from samples of tunnelled wood (n = 20, standardised to 1 litre) within each forest (p = < 0.01) and, (D) the number of species also removed and counted from samples of tunnelled wood (n = 20, standardised to 1 litre) (p = > 0.05): from each of the five mangrove forests (letters above the bars = Tukey’s pairwise post-hoc comparisons, mean ± SE).

The lowest emersion times in the One Onitu, Langira, Loho and Kaluku forests ranged from 7.1 ± 0.5 h per day to 7.7 ± 0.3 h per day (mean ± SE). However, the emersion time recorded in the Gili mangrove was significantly greater compared with all other forests, with a mean emersion time of 12.5 ± 0.2 h per day (Fig. 4B, 1-way ANOVA, F4,156 = 34.3, p = < 0.001, mean ± SE).

The Gili forest had the lowest abundance (6.6 ± 1.6 animals) and number of species per litre volume (3.1 ± 0.7 species) removed from the LWD samples, but Kaluku had the greatest abundance (44.7 ± 16.4 animals) and number of species (11.8 ± 3.6 species) within teredinid-attacked LWD (Figs. 4C and 4D, 1-way ANOVA, F4,95 = 3.75, p = < 0.01 and F4,95 = 2.13, p = > 0.05, mean ± SE respectively).

Temperature measurements within wood were significantly cooler compared with the ambient air temperature outside of the wood (Paired t-test, p = < 0.001). As ambient air temperature increased, internal wood temperatures were significantly cooler, with the greatest difference being −9.5 °C (Fig. 5, Regression analysis R2 = 82%, F1,25 = 116, p = < 0.001) and the abundance of animals within wood increased with cooler temperatures (Regression analysis R2 = 19%, F1,25 = 5.9, p = < 0.05). The cooler the internal temperature within wood relative to ambient air temperature, the greater the abundance of animals found within teredinid-attacked wood (Pearson Correlation, −0.5, p = < 0.01).

Figure 5 Differences of temperature in wood compared with outside air temperatures, and animal abundance.

Temperature difference in-wood compared with outside wood-surface air temperatures (°C) of fallen logs (n = 27) attacked by teredinids in the Langira mangrove forest, with total counts of animals removed from samples of the same logs standardised to one litre. As outside wood-surface air temperatures peak, internal wood-temperature becomes significantly cooler with a maximum temperature difference of >9 °C within wood (p = < 0.001). The abundance of animals also significantly increase with decreasing temperatures (p = < 0.05).

Discussion

The data from this study supports our hypotheses that vacant teredinid tunnels benefit many species in Indonesian mangrove forests, and a greater surface area of tunnels will maintain a greater number of species and abundance of fauna. Animal diversity is positively correlated with increasing habitat complexity (Petren & Case, 1998; Gratwicke & Speight, 2005; Saha, Aditya & Saha, 2009). If it was not for the tunnels created by teredinids, LWD would have significantly fewer animals, and a reduced nursery function, important for species such as octopods, and dartfish (Hendy et al., 2013). Thus, the fauna collected from the teredinid-attacked LWD in the mangrove forests from this study are maintained by a habitat-specific structure—teredinid tunnels inside LWD.

Large pieces of woody debris within aquatic ecosystems are important constituents in the structure and complexity of waterways and will probably become colonised by a wide variety of animals (Hilderbrand et al., 1997; Storry et al., 2006; Hendy et al., 2013). Fallen wood represents a vital resource providing a spatial subsidy for many animals within the aquatic ecosystem (McClain & Barry, 2014). Wood enhances habitat complexity (Shirvell, 1990; Brooks et al., 2004), the deposition of sediments, and the retention of organic matter (Bilby & Likens, 1980; Smock, Metzler & Gladden, 1989). With the removal of wood however, sediment discharge will increase and a reduction of ecosystem-level habitat structural complexity will occur (Larson, Booth & Morley, 2001; Brooks et al., 2004). Large woody debris is therefore an important component within aquatic ecosystems (Shields Jr, Knight & Stofleth, 2006). Thus, a major threat to mangrove ecosystems is wood harvesting (Valiela, Bowen & York, 2001; Duke et al., 2007; Sanchirico & Mumby, 2009) which could reduce ecosystem-level mangrove faunal diversity due to the reduced wood volume (Benke et al., 1985; Wright & Flecker, 2004; Hendy et al., 2013) and lack of niches otherwise created by teredinids.

Indeed, the results from this study corroborate previous research that LWD does increase animal diversity (Everett & Ruiz, 1993; Wright & Flecker, 2004). Teredinids create niches as they consume LWD, by creating tunnels for a wide range of animals—but only when those tunnels become vacant. As the number of teredinid tunnels increase within LWD, the abundance and diversity of animals will become greater. The great amounts of animal abundance and number of species found within increasing numbers of teredinid tunnels may likely be due to a higher proportion of refuges from predation (Willis, Winemiller & Lopez-Fernandez, 2005; Hendy et al., 2013). For example, the refuge provided by woody detritus is exploited by grass shrimp, as LWD significantly reduces their risk of predation from predatory fish (Everett & Ruiz, 1993). Vulnerable species have more options for avoiding and escaping potential predation in habitats containing a greater number of niches. Structurally complex habitats may also reduce visual contact, encounter rates and aggressive behaviour between competitors (Jones, Mandelik & Dayan, 2001; Willis, Winemiller & Lopez-Fernandez, 2005).

Cryptic niches are typically exploited by animals to avoid predation (Ruxton, Sherrett & Speed, 2004), and animals may also exploit LWD to avoid extreme air temperatures. Many areas of the mangrove environment may be affected by rapid fluctuations in temperature (Taylor et al., 2005; Bennett, 2010). Yet, as air temperatures increased, the temperatures measured within LWD became cooler creating a more desirable environment, and the abundance of animals in cooler samples of LWD increased, which may also reduce their risks from desiccation.

Juvenile dartfish, Parioglossus interruptus, although able to tolerate high temperatures, reside in the cooler teredinid tunnels during low tide (Hendy et al., 2013). Under laboratory conditions, emerged LWD has an evaporative cooling process (Hendy et al., 2013). The fauna from this study may also benefit from a lower metabolic rate due to the significantly lower temperatures within teredinid-attacked wood. Evidence of breeding within the tunnels was found, and the cooler internal temperatures in LWD may provide a key refuge for juveniles to escape high temperatures, as well as seeking relative safety from predation.

Temperature is the primary factor affecting development of invertebrates (Smith, Thatje & Hauton, 2013). Previous studies have shown that survival rates of developing veliger gastropods decrease with increasing temperatures due to higher energetic demands of development at higher temperatures (Smith, Thatje & Hauton, 2013). In this study, we found many octopods with egg sacs lining the vacant teredinid tunnels. Octopods in Indonesian mangrove forests may benefit by residing inside cooler teredinid tunnels, as metabolic processes may be slower in cooler temperatures, which may prolong octopod embryonic development—producing stronger hatchlings (Robison, Seibel & Drazen, 2014).

The contribution of teredinid-attacked LWD, and the evaporative cooling within wood (Hendy et al., 2013) to mangrove biodiversity maintenance is significant and remarkable. Mangrove forest biodiversity is significantly enhanced by a large volume of teredinid attacked LWD and the cooler temperatures in LWD may also enhance the development of eggs and juveniles found in the teredinid tunnels. Non-tunnelled LWD has a limited number of species and abundance of individuals due to the reduced niche availability. A lack of teredinid tunnels within LWD maintains a reduced habitat complexity that may likely increase predator–prey encounters. Differences in spatial structure will influence the frequency of interactions such as predation or niche exploitation for animals (Warfe & Barmuta, 2004; Nurminen, Horppila & Pekcan-Hekim, 2007). This may also be the case for the spatial structure teredinid tunnels provide in LWD, which explains the sharp change in animal assemblages and diversity of animals in LWD without tunnels when compared to LWD with tunnels.

Biodiversity is dependent on the substratum sample size (Magurran, 2004), as larger samples are likely to contain additional resources and therefore greater numbers of species. To effectively rule out the factor of sample size from this study all LWD samples were standardised to the same volume and the same number of samples for each mangrove forest were used to test for differences between localities. Even so, a greater amount of teredinid tunnels significantly enhanced the animal abundance and numbers of species within LWD samples of the same volume.

Teredinid tunnelling will also influence faunal diversity at the whole ecosystem level in Indonesian mangrove forests. The lowest overall level of teredinid attacked LWD was recorded in the Gili forest, which also had the lowest counts of animal abundance, number of species and longest recorded emersion time when compared with the other four mangrove localities. Teredinid tunneling is accelerated in mangrove forests with limited emersion times (Robertson & Daniel, 1989; Kohlmeyer, Bedout & Volkmann-Kohlmeyer, 1995; Filho, Tagliaro & Beasley, 2008). Teredinids cannot tolerate regular prolonged emersion such as LWD found in the high intertidal (Robertson, 1990). In the mid- to low- intertidal zones of a Rhizophora-dominated Australian mangrove forest, Robertson (1990) found that half of the original woody mass of large fallen logs was consumed by teredinids within two years, whereas, in the high-intertidal where teredinids were absent, only five percent of the original mass of fallen logs had been lost over two years—explaining the reduced surface area of teredinid tunnels measured within LWD from the Gili mangrove forest. This means that the biggest effects are attributable to teredinid communities living at greater densities, such as that found with the Kaluku mangrove forest, which had the highest degree of teredinid tunnelling in LWD and the greatest abundance of animals and number of species within those tunnels. Although LWD is essential for the biodiversity of both the specialist and more generalist animals (Hilderbrand et al., 1997; Kappes et al., 2009; Hendy et al., 2013); teredinid tunnels will increase the internal structural complexity within LWD and the tunnels significantly enhance biodiversity.

Teredinid tunnels are home to many vulnerable animals (juveniles and adults), which cannot bore into the very hard, un-decayed wood. Thus, the large numbers of animals that rely on teredinid tunnels for predation refugia, or environmental buffering, or both, would not be as abundant, or may not even be present in the mangrove ecosystem if it were not for the tunnelled wood. Based on the data presented here we classify teredinids as ecosystem engineers. A critical characteristic of ecosystem engineering is that the engineered modifications, in this case teredinid tunnels, must change the availability (quality, quantity and distribution) of resources utilised by other fauna (Jones, Lawton & Shachak, 1994). Vacant teredinid tunnels within LWD in mangrove forests provide many niches and the high complexity of tunnels lead to a broad range of co-existing animals within LWD, especially in LWD with significantly greater surface area of tunnels. Notwithstanding, the considerable turnover of large volumes of fallen wood by teredinids in mangrove habitats—as the processed wood coupled with teredinid tissue and faecal matter may significantly contribute to mangrove out-welling of nitrogen and carbon, improving the productivity of near-shore adjacent ecosystems.

Spatial heterogeneity is a fundamental property of the natural world (Kostylev et al., 2005) and heterogeneity within an ecosystem is a vital component for the interaction of co-existing animals (Petren & Case, 1998; Gratwicke & Speight, 2005). Increasing habitat complexity may reduce trophic interactions and subsequently increase ecosystem stability (Kovalenko, Thomaz & Warfe, 2012). By comparison structurally simple habitats are not able to support the same levels of biodiversity when compared with habitats consisting of high levels of complexity and rugosity (Levin, 1992). If mangrove harvesting and wood removal persists, then Indonesian mangrove faunal abundance and diversity will be significantly reduced due to the lack of tunnel niches created by teredinids.

Supplemental Information

Supplemental Information 1 Raw data with numbers of abundance/species from each mangrove forest

Total data from all LWD samples collected from each of the five transects within each mangrove forest locality.

Click here for additional data file.

Supplemental Information 2 Regression and analysis of means of randomly selected LWD categorised by the percentage surface area of teredinid tunnels

Raw data of LWD categorised by the percentage surface area of teredinid tunnels, with corresponding abundance of and numbers of species.

Click here for additional data file.

Supplemental Information 3 Inside wood and outside air temperatures, with animal abundance

Raw data of internal wood (n = 27) and outside air temperatures, and the abundance of animals removed from the same samples of wood.

Click here for additional data file.

We thank D Smith, P Mansell, and T Coles for support during field activities. We also thank Amat and Kundang, for their hard work and help with fauna collections. We also send our sincere gratitude to RSK Barnes for help with the identification of fauna. Finally, all authors of this manuscript wish to give special thanks to each of the reviewers for their expert comments and suggestions.

Additional Information and Declarations

Competing Interests

Author Contributions

Field Study Permissions

The authors declare there are no competing interests.

Ian W. Hendy conceived and designed the experiments, performed the experiments, analyzed the data, contributed reagents/materials/analysis tools, wrote the paper, prepared figures and/or tables, reviewed drafts of the paper.

Laura Michie and Ben W. Taylor performed the experiments, contributed reagents/materials/analysis tools, reviewed drafts of the paper.

The following information was supplied relating to field study approvals (i.e., approving body and any reference numbers):

Field study was agreed and approved by Operation Wallacea under permit 04/TKPIPA/FRP/SM/IV/2011.

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
