# Peer review of "Habitat creation and biodiversity maintenance in mangrove forests: teredinid bivalves as ecosystem engineers"

_PeerJ, doi:10.7717/peerj.591_

## Round 0.1 · original submission · Minor Revisions

Please pay particular attention to the statistical ambiguities raised by reviewer 2; provide clarity to your tables and figures as suggested by both reviewers; and ensure clarity by doing a thorough English language edit.

·

Basic reporting

This is a nice, neat idea for a paper and there are some interesting findings that emerge. The literature is adequately covered and the presentation style is appropriate but does include some awkward wording.

Experimental design

No real problems with this, there is adequate replication both within and between sites

Validity of the findings

Findings are robust and interesting trends are found. The presentation style is a bit lacking in several places and I have made quite a few suggestions for minor corrections, but if these are carried out the paper is acceptable

Additional comments

There were quite a few style and presentation errors in this paper which i have noted below by line number:
Line 50 few species and LOW abundance ( cannot have 'few abundance')
L 87 into is one word ( see also L 106)
L 89 not a 'wide number' - a wide range of a large number!
L 138 and TRANSECTS were between 50-100 m apart
L144 'and were expressed as numbers per litre of wood'
L160 and other places - be consistent with giving numbers as numerals or a words. Here we have 2 probes, but earlier five forests ( and later again 5 forests!). Numbers up to nine are conventionally given as words, those over 10 as numbers.
L165 laboratory not lab
L211 juveniles of many species ( there is no such thing as a juvenile species)
L212-213 This sentence is badly written and makes no sense - do you mean something like: One variable, percentage surface area of teredinid attach, was responsible for explaining xx % of the difference between animals abundance and numbers of species... Also in this section surely we are interested in what factors explain DIFFERENCES rather than similarities - i would have turned this around and given the % contribution to difference?
L222 Another awkward sentence here - i think you mean to say: No significant correlations were found between abundance of animals on wood and either salinity, distance form and or forest area ?
L235 Forest cannot be' statistically indistinguishable' must mean some characteristics of those forests?
L 244 'measurements was' - you are mixing pleural and singular here
L 254 I am sure 'you support' your hypothesis, but you should be stating whether the data support it!
L 264 waterways ( one word)
L 329 enhanced not enhance
L344 badly expressed - you mean something like 'Based on the data presented here we classify teridinids as.....

I do not think Fig 1 images add value to the paper and suggest this be removed
I would like to see a full species list of the taxa found in each forest, instead of just the groups, in Table 2

·

Basic reporting

While this article does place the current study within the context of other work considering mangroves, it does not adequately utilise previous work on habitat complexity and its relationship to diversity. In the in-text comments I have suggested numerous papers that should be considered.
.
Figures 4 and 5 are not clear, especially Figure 5. I have provided detailed feedback in my in-text comments.

Experimental design

There are methods that need to be clarified (e.g. how the samples were converted to a measure of volume). Detailed comments have been made in the in-text comments.

Validity of the findings

I have a few concerns related to the statistical analyses employed:
• A different number of samples were taken at the different sites and then used to compare diversity. This is problematic as the more samples taken the more species one would expect to be recorded. Suggestions on how to correct this can be found in the in-text comments.
• The reporting of statistical findings is ambiguous throughout, and in some places contradictory.
• Correlations are employed but then reported as cause and effect.
• As the number of animals and species have been recorded I suggest using a diversity index such as Shannon-Wiener to compare diversity between sites rather than just using species number.

Additional comments

This work is a nice contribution to mangrove ecology and particularly to marine ecology in Indonesia. I encourage you to address the comments and pursue publication.

---

## Round 0.2 · Minor Revisions

I am still picking up some grammatical and spelling errors in the manuscript which needs to be carefully proof read.
These include the following suggested edits (not exhaustive):
Line 98-99: therefore the diversity of animals and the abundance of individuals are largely considered to be attributed to deterministic....
Line 103: Although mangrove roots provide a nursery habitat...
Line 56: The percentage of teredinid tunnels
Line 256: greater volumes
Line 261-262: Also corroborated by a General Linear Model (p = < 0.001), with site and percentage surface area of teredinid attack as factors [This is an incomplete sentence, please re-phrase]
Line 262: The lowest number of species was found
Line 264: The greatest number of species was
Line 284: Rephrase to: The lowest emersion times in the One Onitu, Langira, Loho and Kaluku forests ranged from 7.1 ± 0.5 hours per day to 7.7 ± 0.3 hours per day (mean ± SE).
Line 289: The Gili forest had the lowest abundance
Line 300: the greater the abundance of animals found within
Line 313: Large woody debris within aquatic ecosystems is an important constituent
Line 370: Differences in spatial structure
Line 385: and longest recorded
Line 386: tunneling

---

## Round 0.3 · accepted · Accept

Thank you for your final revision.